# Factors associated with unintended pregnancy and contraceptive practices in justice-involved adolescent girls in Australia

Helene Smith[1]*, Mandy Wilson[1], Basil Donovan[2], Jocelyn Jones[3], Tony Butler[1], Sally Nathan[1], Paul Simpson[1]

1 School of Population Health, University of New South Wales, Sydney, Australia, 2 Kirby Institute, University of New South Wales, Sydney, Australia, 3 National Drug & Research Institute, Curtin University, Perth, Australia

☯ These authors contributed equally to this work.
* Helene.smith@unsw.edu.au

## Abstract

### Introduction

Despite a decline in unintended teenage pregnancy in Australia, rates remain higher amongst justice-involved adolescent girls, who are more likely to be from disadvantaged socio-economic backgrounds, have histories of abuse, substance use and/or mental health issues. Furthermore, exposure to the criminal justice system may alter access to education and employment and opportunities, potentially resulting in distinct risk-factor profiles. We examine factors associated with unintended pregnancy, non-contraceptive use and Long-Acting Reversible Contraception (LARC) in a sample of sexually active, justice-involved adolescent girls from Western Australia and Queensland.

### Methods

Data from the Mental Health, Sexual Health and Reproductive Health of Young People in Contact with the Criminal Justice System (MeH-JOSH) Study was analysed on 118 sexually active adolescent girls. Participants were aged between 14 and 17 years, purposefully sampled based on justice-system involvement and completed an anonymous telephone survey. We constructed two multivariate models taking reproductive outcomes as the dependent variables.

### Results

Over one quarter (26%, 30/118) reported a past unintended pregnancy, 54 did not use any contraception at their last sexual encounter, and 17 reported LARC use. Following adjustments in the multivariate analysis, lifetime ecstasy use was associated with both unintended pregnancy (aOR 3.795, p = 0.022) and non-contraception use (aOR 4.562, p = 0.004). A history of physical abuse was also associated with both any contraception (aOR 3.024, p = 0.041) and LARC use (aOR 4.892, p = 0.050). Identifying as Aboriginal & Torres Strait Islander, education/employment status and geographic location appeared to have no association.

**Data Availability Statement:** We are unable to make the data available publicly for a number for reasons related to the ethical restrictions placed by our ethics research committee when providing

consent for this study. The reasons for this are outlined below: i. The data set contains highly sensitive information about participants' sexual history, past substance use, medical history (including HIV status, history of mental health), criminal activity (such as illegal drug use and paid sex work), and reproductive health history (including history of terminating pregnancies). In addition, it includes histories of sexual and physical assault, which may or may not have been previously disclosed to the police. Linking of any of these issues to a specific participant could be highly detrimental to the physical and mental wellbeing of the participant, as potentially expose the participant to both stigma and/or potential legal complications. ii. Whilst all data has been de-identified, the relatively small sample size (118), combined with geographical information, details of criminal history, age of sexual partners and other identifying information, could make a participant identifiable, or (just as damagingly) have an individual be perceived to be a specific participant who has engaged in certain stigmatising behaviour. Given the high level of vulnerability of this cohort, many of which may be waiting on sentencing, parole orders, custody arrangements or other interactions with government and justice bodies, we do not believe the risk of disclosure to be acceptable by making such a dataset publicly available. iii. As part of the consent process, participants were assured that their data would remain private and only be made available with the express permission of the investigator team where we could ensure the ongoing privacy of the data. We therefore believe that making this data (even if de-identified) publicly available would be a breach of our agreement with participants. We are, however, able to provide the data set for review by individuals at PLOS ONE upon request. The contact details for the ethics committee are listed below: NSW Human Research Ethics Committee HREC Ref: # HC13308 humanethics@unsw.edu.au 02 9348 1943.

**Funding:** Tony Butler Grant No. 1043693 National Health and Medical Research Council (NHMRC) Project he funders had no role in study design, data collection and analysis, decision to publish, or preparation of the manuscript.

**Competing interests:** The authors have declared that no competing interests exist.

## Conclusion

Our findings suggest that justice-involved adolescent girls have distinct risk factors associated with unplanned pregnancy and contraception use compared to the general population, but more research is required to understand the mechanisms and contexts underlying these risk factors. How exposure to physical violence may encourage contraception and LARC use, in particular, warrants further attention as does the association with ecstasy use.

## Introduction

Teenage pregnancies are associated with poor health outcomes for both the mother and child globally. Studies from high-income countries comparing the health outcomes of pregnant adolescents to adults have found higher rates of still birth, pre-term delivery, low-birth weight, neonatal mortality, congenital anomalies [1–3] and lower offspring IQ [4]. These outcomes are more pronounced where the maternal age is less than 16 years [5] and are frequently influenced by the behavioural, economic and social factors that may place adolescent girls at a higher risk of pregnancy [5]. Whilst adolescent mothers will frequently view motherhood as a positive experience which may elicit aspirations related to educational and career attainment [6, 7], social stigma and economic barriers often result in such young mothers being more likely to experience increased economic disadvantage [8], lower levels of education and employment [9], and be at increased risk of post-partum depression [10], although these findings are often moderated by socio-economic status [9, 10]. Unintended pregnancy within any given population has recently been identified as a proxy indicator for measuring women's access to health services and level of reproductive autonomy [11].

In Australia, the number of teenage pregnancies in the general population has declined over the past 30 years [12, 13]. This has largely been attributed to a higher uptake in contraception, with user-dependent methods such as the contraceptive pill and condoms the most commonly used [14]. User-independent methods, that require no actions on from the user after initial insertion, such as implants and intrauterine devices (IUDs), known as long-acting reversible (LARC), contraception, are increasingly recommended as first-line contraception for women of all ages, including adolescents, regardless of pregnancy history, by a number of advisory bodies in Australia [15–17]. This is driven by their near perfect use failure rates [18, 19], an important factor for adolescent populations who report higher levels of contraceptive failure when using user-dependent methods compared to older women [20, 21]. In addition, LARC use has been reported as highly acceptable amongst adolescent users who value the convenience, privacy this method offers [22].

Despite this, uptake remains low in the general population [23] and even more so amongst adolescent girls (<2%) [14]. This is in part driven by concerns around side-effects [24], apprehension around the insertion process [24, 25], and fears related to having a foreign object inserted into the body [26, 27].

Furthermore, issues of cost and accessibility have also hampered uptake of LARC, particularly when looking at the availability of trained clinicians to perform insertions [28, 29]. Whilst the costs of insertions and removals performed by GPs and medical specialists are covered by the Australian Medicare Benefits Schedule, registered and private nurses may struggle to access–increasing costs to the consumer, particularly in areas with limited access to GPs [30]. Furthermore, GPs have argued that the rebate currently available does not adequately represent the costs involved in performing the procedure (including associated training costs), resulting in many GPs declining to offer the service [31].

In addition, several questions remain around LARC's universal appropriateness, particularly when considered in the context of the economic and cultural framework in which women and girls who have had contact with or interaction with the criminal justice system, such as the police, judicial or correctional systems (ie. justice-involved women and girls) may make decisions around contraception [32]. Some scholars have pointed to its use as a form of soft sterilisation and social control, particularly when financial motivations or coercive counselling lead to women and girls having devices inserted which they are unlikely to be able to have easily removed to due access issues [33–36]. The degree to which this impacts contraceptive decision making within the Australian correctional system remains under-explored however the phenomena is well documented with the US Correctional system [37, 38]. These issues are compounded by reports of racial profiling by healthcare providers when offering contraception options [39–44] as well as women & girls with abuse histories being less likely to be comfortable with IUDs and other vaginally inserted contraception devices [45, 46]. Within this context, the need for shared and informed decision making between provider and patient is crucial [47] and understanding factors that may be associated with use and non-use within a justice involved Australian population are needed.

Teenage pregnancies in Australia are mostly unintended and nearly half are terminated [13]. Higher rates of teenage pregnancy continue to be reported amongst adolescent girls with specific socio-demographic characteristics. These include adolescent girls who live in rural, remote and/or socio-economically disadvantaged areas [12], Aboriginal & Torres Strait Islander women [12, 48], adolescent girls born to adolescent mothers [49, 50] and adolescent girls who have been to exposed to either family violence or sexual abuse [51]. The reasons for this are complex and have been attributed to a mix of limited access to culturally appropriate services [48], reticence to discuss reproductive health needs with healthcare staff [52],, poor contraception negotiation skills with partner [53–55], disengagement from secondary school [56, 57], low-risk perceptions and general ambivalence towards pregnancy [58, 59], as well as exposure to reproductive coercion [48, 60]. Reproductive coercion refers where a person is where a person is exposed to behaviour that interferes with their ability to make autonomous decisions about their reproductive health [61]. This could involve an individual, such as a partner or family member who may sabotage or force contraception use, or alternatively force a women to either terminate or keep a pregnancy against her wishes. It can also occur at a structural and systematic level when institutions and/or individuals in a position of power place restrictions and controls on women's reproductive freedom [62]. It can be as blatant as legislation and/or policy that places limits on access to reproductive health services or as insidious as a healthcare or social worker making a recommendation on whether or not an individual can access certain contraceptive options based on personal judgments about the client's personal circumstances [62].

Justice-involved adolescent girls aged between 14 and 17 years, defined as girls who have interacted with the justice system through detention, community orders or have been subjected to repeated warnings, cautions and fines, have also been identified as a group who report high levels of unintended pregnancies and terminations compared to their counterparts in the general population [63, 64]. Whilst adolescent girls only represent about 6% of all young people sentenced to detention, they make up 19% of young people remanded in custody, 16% under community based supervision orders and 23% are subjects of a criminal justice case [65]. Routine surveys, both in Australia and oversees, have found that they are also more likely to be from disadvantaged socio-economic backgrounds [12, 66], engage in high-risk sexual behaviours [13, 66, 67] and have a history of sexual and physical abuse, mental health problems and substance use [68]. Substance use has been shown to be a risk factor for both risky sexual behaviours (such as sex while intoxicated, or condomless sex with a casual partner,

transactional sex, etc.) and being exposed to non-consensual sex or intimate partner violence [69–76], placing them at a higher risk of unintended pregnancy.

Considerable work has been done to examine the relationship between characteristics commonly found in juvenile justice populations and risky sexual behaviour, particularly in the United States. Factors include gang membership [77], substance use [78] age of sexual debut [77] and partner communication [79]. Minimal work however has been completed to understand how these risk factors may interact to impact both contraception use and unintended pregnancy [80] and no studies have been published in Australia, which has a distinct demographic and cultural incarceration profile compared to juvenile adolescent girls with a history of incarceration in the United States. This includes the over-representation of Aboriginal & Torres Strait Islander people who currently make-up 51% of juvenile detainees in Australia despite being 3% in the community [81], with the number of Aboriginal & Torres Strait Islander adolescent girls appearing in court steadily increasing over the last four years [82]. Aboriginal & Torres Strait Islander women have also been shown to have distinctive contraception use patterns compared to women in the general Australian population, including lower use of any contraception, but higher uptake of LARC [83]. Qualitative interviews with Aboriginal women from remote communities have revealed lower levels of stigmatising views around teenage pregnancy compared to the general population [60, 84] but have also reported high levels of reproductive coercion [48, 60, 84]. Understanding which risk factors emerge as significant is this population mix therefore merits further investigation.

The Mental Health, Sexual Health and Reproductive Health of Young People in Contact with the Criminal Justice System (MeH-JOSH) survey was conducted amongst a sample of adolescents aged 14–17 years, from Western Australia and Queensland, who had a history of involvement in the criminal justice system [63, 85]. Amongst adolescent girls, the study found considerably higher rates of pregnancy [86] compared to findings from similar adolescent samples in the general community [14]. The study also reported very high rates of miscarriage and stillbirth as well as engaging in health risk behaviours (tobacco smoking, drinking alcohol and taking illicit drugs) while pregnant [86].

We created a subset of this database to conduct a multi-variate analysis of factors associated with unintended pregnancy, non-contraceptive use and LARC uptake amongst sexually active girls. Understanding which factors are associated with these outcomes may allow a greater focus on key areas of intervention and support.

## Methods

### Study design

A cross-sectional survey was administered to all participants where they were asked questions about their past involvement with the police and/or incarceration history, demographic information (including whether they identified as Aboriginal and/or Torres Strait Islander), contraception use and pregnancy history, history of sexual and/or physical abuse, mental health, substance use, sexual identity, history or sexual activity and sexual health, as well as any self-reported cognitive issues or past history of self-harm, suicide attempts and head-injuries.

### Sampling & sample size

Participants for the MeH-JOSH study were recruited exclusively from the community in the Australian states of Queensland and Western Australia. No participants were recircuited that were being actively detained or serving a sentence. A purposive sampling design was used to recruit young people who met the eligibility criteria. Participants needed to be aged between 14 and 17 years and report either current or past contact with the criminal justice system [63].

For this analysis, only participants that were female and had reported past sexual activity with a male, defined here as penetrative vaginal sex, were included in the analysis.

Quota sample sizes for recruitment were calculated based on known demographic characteristics of the Australian juvenile-involved population [87]. This included oversampling of Aboriginal & Torres strait Islander girls, who are over-represented in the justice system [88].

## Recruitment

Recruitment took place between June 2016 and August 2018. To minimise selection bias, a number of recruitment strategies were applied. In both states, young people who met the selection criteria were recruited through referrals by programme coordinators from community-based organisations and youth drop-in centres as well as young people attending flexi-learning schools or colleges, institutions which aim to re-engage young people who have been disengaged from formal schooling. In Queensland, young people were also recruited whilst waiting either inside or outside magistrates' courts on days the Children's Court was in session. This strategy was not applied in Western Australia due to not being granted permission to reimburse participants who were attending court. Programme coordinators then referred young people who satisfied the selection criteria and agreed to participate in the study. All participants were provided with a $50 cash gift voucher to compensate for their time as well as a bag which contained chocolates, drinks and muesli bars.

## Consent

Human research ethics approval was received for young people recruited in the survey to be treated as mature minors and consequently, waived the need for parent and/or guardian consent [89] with recruiters administering a Gillick Competency [90] checklist for participants aged <16 years of age to ensure they met the criteria of a mature minor. Following confirmation as a mature minor, participants aged under 16 years were asked by the research team for permission to contact a parent or guardian to obtain parental consent. If either the young person refused to provide contact details or the parent/guardian refused consent, the young person was still able to participate, having been identified as a mature minor.

## Survey

After having received written consent from the participant, the survey was delivered using a Computer Assisted Telephone Interview (CATI). The average time to complete the survey was approximately 40 minutes. No identifying information was recorded by the interviewer to ensure anonymity.

Following participation in the survey, participants took part in a debrief interview and, if necessary, were provided with referrals to relevant health agencies to ensure they were not adversely impacted by the survey questions.

Data were collected on the following: socio-demographics; history of justice system involvement; sexual identity and sexual history; sexual health behaviours and knowledge; past history of pregnancy, outcomes associated with pregnancy and high-risk behaviours during pregnancy; current contraceptive method use; past history of engagement with clinical health services and diagnosis of sexually transmissible infections (STIs) and a history of head injury (with or without a loss of consciousness). In addition, participants were screened for indications of abnormal prosocial behaviour and psychopathology using the Strengths & Difficulties Questionnaire (SDQ) [91], history of mental health issues including major depression, Attention-Deficit/Hyperactivity Disorder (ADHD) and psychosis using the M.I.N.I International Neuropsychiatric Interview for Children and Adolescents 6.0 (MINI Kid) [92], as well as any

history of self-harm and/or suicide attempt. Participants were also screened for current alcohol use and dependence using the MINI Kid [92] as well as past use of a number of illicit substances. Finally, participants were screened for a history of sexual and physical abuse, as well as social control. Sexual abuse was determined by answering "yes" to the question *"In your lifetime, have you ever had sex when you didn't want to*?", physical abuse was determined by answering "yes" to *"In your lifetime, did someone ever try to physically hurt you (e.g. hit, slap or kick you)*?", and social control was determined by answering "yes" to *"In your lifetime, did someone ever try to limit your contact with family or friends)*?". Survey questions were designed to be consistent with other Australian surveys of young people to enable comparisons. Survey questions are provided in the supplementary material (S1 File).

## Outcomes of interest

Three outcomes of interest were examined. Firstly, unintended pregnancy was defined as any participant who reported being pregnant in the past and when asked whether or not the first pregnancy was planned, responded no. Secondly, non-contraceptive use was defined as any participant who, when asked whether nor not they used contraception at their last sexual encounter, reported that they used no form of modern contraception. We defined "modern contraception" according to the definition outlined by Hubacher in 2015 [93] and generally applied within the literature [94–96] as any *"product or medical procedure that interferes with reproduction from acts of sexual intercourse"*. This included condoms, pills, patches, vaginal rings, injections, IUDs and implants. Participants who responded that they used the withdrawal method were considered non-contraception users. Thirdly, LARC use was defined as any participant who, when asked to specify what contraception they used at their last sexual encounter, specified using a LARC method including implants and IUDs.

## Data analysis

Analysis was conducted using Statistics SPSS 27 IBM SPSS Statistics, Armonk, New York. We constructed two multivariate models taking reproductive outcomes as the dependent variable (unintended pregnancy, non-use of contraception and LARC use) and demographics, as well as other risk factors (criminal history, abuse history, mental health, other cognitive considerations, substance use, sexual identity, history of sexual activity and sexual health as well as current sexual activity) as independent variables.

## Independent variable selection

Two models were developed. One which examined independent associations with unintended pregnancy and another which examined independent associations with contraceptive practices (both non-contraceptive use and LARC use). A univariate analysis was conducted on all independent variables. Variables that showed a significance of p<0.2 across any of the dependent variables were then added to the multivariate models together with key demographic variables. As the contraceptive use outcomes were more likely to relate to current behaviour as opposed to past behaviour, as may be the case with the unintended pregnancy outcome, a number of conceptual decisions were made in relation to variable selection compared to the final model for unintended pregnancy. For example, in the model examining contraception-use, associations with current depression were examined whereas in the unintended pregnancy model, we examined associations with a past history of a depressive episode. Otherwise, the same variables were considered across both models.

Variable selection was informed by a mix of theoretical concerns and/or previous studies that have shown associations with the variables of interest. This included examining

demographic factors such as geographic location [12], Aboriginal &/or Torres Strait Islander identity [12, 48], employment status [97] and sexual identity [98]; abuse histories including sexual and physical assault [51], as well as reporting first time sex with significantly older partners (> 5 years) [99]; history of mental health issues including depression [100] and PTSD [101],; sexual health knowledge scores [102] and past history of illegal substance use [103].

When considering which illegal substances to include, we excluded cannabis use, due to the extraordinarily high numbers reporting use (87%, 103/118) and included ecstasy as it was the second highest most cited substance, highly correlated with other drug use such as methamphetamines, LSD, amphetamines. These other substances, as well alcohol dependence or abuse were excluded as initial modelling with these variables showed no effect.

We also excluded a number of variables from model consideration that were more likely to occur following becoming pregnant. Specifically, engagement with clinical services, as well as STI testing and diagnosis, which are routinely offered to any woman under 30 who attends a clinic visit related to pregnancy [104]. In addition, we excluded the ADHD variable given clinical recommendations that ADHD should not be diagnosed in adolescents with high trauma histories (as was the case with this group) without further clinical investigations [105].

## Ethics

Approval for the study was obtained from the University of New South Wales Sydney Human Research Ethics Committee, (HC13308), the Western Australia Aboriginal Health Ethics Committee (WAAHEC 625), and Curtin University (HRE0133). Permission was also granted in Western Australia by the North Metropolitan Health Service Mental Health Research Ethics Committee (22_2016), the North Metropolitan and East Metropolitan Health Services' Research Governance, and the Department of the Justice Research Application Advisory Committee (ref 2016/02161) to recruit from their respective premises.

## Results

### Participant characteristics

All participant characteristics are outlined in Table 1 below.

**Demographic profile.** A total of 118 participants aged 14 to 17 years, with a history of past vaginal intercourse with a male, were identified from the MeH-JOSH dataset. Eighty percent (94/118) were currently living in an urban setting (metropolitan Perth or Brisbane), 90% (106/1180 were born in Australia and 42% (49/118) identified as Aboriginal and/or Torres Strait Islander (First Nations or Indigenous).

**Abuse & mental health.** One third of participants (33%, 39/118) reported a history of unwanted sex due to partner pressure or coercion, 62% (71/118) reported a history of physical abuse and almost half (47%, 56/118) reported someone attempting to limit their contact with family or friends. Eighty-six percent (101/118) reported either a past or current history of depression. Just under two-thirds (63%, 74/118) reported a history of self-harm and/or 47% (55/118) reported ever attempting suicide (Table 1).

**Sexual identity, history of sexual activity and sexual health.** Nearly a quarter (22%, 26/118) identified as bisexual, lesbian, queer or homosexual. Fifteen percent reported their first sexual encounter before the age of 13, and 11% stated that their first sexual partner was five or more years older than them. (Table 1).

**Substance use.** Median age for both first alcohol and illicit drug consumption was 13 years. Forty-one percent (55/118) were identified as likely to suffer from current alcohol dependence or abuse, and 91% (107/118) reported past illicit drug use.

**Table 1. Participant characteristics.**

| | | N | % |
|---|---|---|---|
| Demographics | Median age | 16.0 years | |
| | Recruited in an urban setting | 94 | 80% |
| | Aboriginal & Torres Strait Islander | 49 | 42% |
| | Born in Australia | 106 | 90% |
| | Not enrolled in education or employment | 36 | 31% |
| Criminal history | History of serious sentence–history of juvenile detention, police watch or adult prison | 38 | 32% |
| | TYPE OF OFFENCES PREVIOUSLY COMMITTED<br>Theft (includes break and enter, robbery) | 61 | 52% |
| | Assault (includes sexual & physical injury and/or harassment) | 40 | 34% |
| | Drug Charges | 10 | 8% |
| | Public order offenses | 15 | 13% |
| Abuse history (inflicted on the participant) | Forced or frightened to have sex | 39 | 33% |
| | Physical abuse | 71 | 60% |
| | Social control | 56 | 47% |
| Mental health | Any current or past episode of depression | 101 | 86% |
| | Past history of depressive episode | 71 | 60% |
| | Recurrent depression | 56 | 48% |
| | Number of depressive episodes (median) | 5 episodes | |
| | Current depression | 64 | 54% |
| | 3 or more psychotic symptoms in the last 12 months | 22 | 19% |
| | Current PTSD | 34 | 29% |
| | History of self-harm | 74 | 63% |
| | Age of first attempt at self-harm (median) | 13 years | |
| | Suicide attempt ever | 55 | 47% |
| Other considerations | Past Head Injury | 42 | 36% |
| | Abnormal SDQ scores ($\geq 20$) | 59 | 50% |
| | Scores on sexual health knowledge test | 92 | 78% |
| Sexual identity, history of sexual activity & sexual health | Queer, Bi-sexual, Lesbian or other non-heterosexual identity | 26 | 22% |
| | Age of first sexual encounter (median) | 14 years | |
| | First sexual encounter <13 years of age | 18 | 15% |
| | Age of first sexual partner (median) | 16 years | |
| | First sexual partner 5 years older or more | 13 | 11% |
| | Has previously sought sexual health advice from a clinical professional | 37 | 32% |
| | Ever tested for an STI | 61 | 52% |
| | Ever diagnosed with an STI | 14 | 12% |
| | Penetrative sex with a male in the last 12 months | 111 | 94% |
| | Penetrative sex with a male at last sexual encounter | 115 | 97% |
| | Last penetrative sexual encounter was with a regular partner | 64 | 55% |
| | Median age of last sexual partner | 17 years | |
| | Last sexual partner age difference $\geq 5$ years | 11 | 9% |
| | Drunk or high at last sexual encounter | 40 | 34% |

(*Continued*)

**Table 1.** (Continued)

| | | N | % |
|---|---|---|---|
| Substance use | Age of first alcohol consumption (median) | 13 years | |
| | Age of first illicit drug use (median) | 13 years | |
| | Current Alcohol Dependence or abuse | 55 | 47% |
| | Ever used illegal substances | 107 | 91% |
| | Injecting drug use | 13 | 11% |
| | Cannabis use | 103 | 87% |
| | Ecstasy use | 41 | 35% |
| | Methamphetamine use | 36 | 31% |
| | LSD &/or other hallucinogens | 22 | 19% |
| | Solvents use | 18 | 15% |
| | Amphetamines use | 17 | 14% |
| | Heroin &/or other opiates use | 13 | 11% |

**Other considerations.** Thirty-six percent (42/118) reported a history of a head injury that had resulted in a loss of consciousness, 50% (59/118) were in the abnormal range (score $\geq 20$) for indications of prosocial behaviour and psychopathology, and 78% (92/118) scored poorly ($<8/12$) on general knowledge questions related to sexual health (Table 1).

**History of pregnancy and contraception use.** Table 2 outlines participant pregnancy history and contraception use at their last sexual encounter. Just over a quarter (26%, 30/118), reported ever having been pregnant, with nearly all (90%, 27/30) reported as an unintended pregnancy. Only a small number (13%, 4/30) of pregnancies ended with a live birth. Well over half (67%, 20/30) reported a miscarriage or stillbirth, and over a quarter (27%, 8/30) terminated the pregnancy.

## Factors associated with unintended pregnancy and contraceptive use

**Unintended pregnancy.** Thirty girls (26%) reported a history of at least one pregnancy of which 93% were described as unintended. Participants with a history of pregnancy (n = 30) reported a median of one pregnancy and a median of 15 years of age at the first pregnancy.

**Table 2. Pregnancy and contraception use.**

| | | N | % |
|---|---|---|---|
| Pregnancy history | Has ever been pregnant | 30 | 26% |
| | Age of first pregnancy | 15 years | |
| | Median number of pregnancies | 1 | |
| | Unintended pregnancy | 27 | 90% |
| Pregnancy outcomes | Live births or still pregnant | 4 | 13% |
| | Terminated a pregnancy | 8 | 27% |
| | Miscarriage | 18 | 60% |
| | Stillbirth | 2 | 7% |
| Reported contraception method at last sexual encounter[a] | Yes (any) | 64 | 54% |
| | Condoms | 46 | 39% |
| | Birth control pills | 8 | 7% |
| | LARC | 17 | 14% |
| | Withdrawal | 2 | 2% |

[a]Multiple responses allowed

**Table 3. Factors associated with a history of unplanned pregnancy among justice involved girls aged 14–17 years (n = 118).**

| Variable | | Univariable OR (95% CI) | Reported P-value | Multivariable OR (95% CI) | Reported P-value |
|---|---|---|---|---|---|
| Demographics | Location | | | | |
| | Urban | 1.0 | | 1.0 | |
| | Regional | 0.8 (0.3–2.4) | 0.7 | 1.1 (0.3–4.2) | 0.9 |
| | Aboriginal & Torres Strait Islander | | | | |
| | Yes | 1.0 | | 1.0 | |
| | N No | 0.9 (0.4–2.2) | 0.9 | 0.8 (0.3–2.4) | 0.7 |
| | Unemployed or permanent disability | | | | |
| | No | 1.0 | | 1.0 | |
| | Yes | 2.0 (0.8–5.0) | 0.1 | 2.3 (0.8–6.9) | 0.1 |
| | Queer, Bi-sexual, Lesbian or other non-heterosexual identity | | | | |
| | No | 1.0 | | 1.0 | |
| | Yes | 2.7 (1.0–7.4) | 0.05 | 2.7 (0.8–9.8) | 0.1 |
| Abuse & sexual histories | Forced or frightened into sex | | | | |
| | No | 1.0 | | 1.0 | |
| | Yes | 2.6 (1.1–6.2) | 0.03 | 3.2 (0.9–11.9) | 0.1 |
| | Physical abuse | | | | |
| | No | 1.0 | | 1.0 | |
| | Yes | 1.3 (0.6–3.3) | 0.5 | 0.4 (0.1–1.3) | 0.1 |
| | Age difference at first sexual encounter ≥ 5 years | | | | |
| | No | 1.0 | | 1.0 | |
| | Yes | 2.3 (0.7–8.0) | 0.2 | 0.6 (0.1–3.1) | 0.5 |
| Mental Health | PTSD | | | | |
| | No | 1.0 | | 1.0 | |
| | Yes | 3.1 (1.2–7.6) | 0.02 | 3.6 (1.1–12.3) | 0.04 |
| | Past history of depressive episode | | | | |
| | No | 1.0 | | 1.0 | |
| | Yes | 1.9 (0.7–4.7) | 0.2 | 1.0 (0.3–3.5) | 1.0 |
| | Suicide attempt ever | | | | |
| | No | 1.0 | | 1.0 | |
| | Yes | 2.5 (1.0–6.2) | 0.04 | 1.1 (0.3–3.8) | 0.8 |
| Substance Use | Ecstasy Use | | | | |
| | No | 1.0 | | 1.0 | |
| | Yes | 3.4 (1.4–8.2) | 0.008 | 3.9 (1.2–12.5) | 0.02 |
| Other Factors | Scores on sexual health knowledge test | | | | |
| | High | 1.0 | | 1.0 | |
| | Low | 0.7 (0.3–2.0) | 0.5 | 0.9 (0.2–3.1) | 0.8 |

OR, odds ratio; CI, confidence interval

In the univariate analysis, factors significantly associated with unintended pregnancy included: past ecstasy use (OR 3.35, p = 0.008), screening positive for current PTSD (OR 3.08, p = 0.015), a history of being forced or frightened into having sex with a partner (OR 2.6, p = 0.032), ever having attempted suicide (OR 2.54, p = 0.040), and identifying as queer, bisexual or lesbian (OR 2.72, p = 0.05). Following adjustment in the multivariable analysis, only lifetime ecstasy use (aOR 3.795, p = 0.022) and PTSD (aOR 3.381, p = 0.046) remained significant (Table 3).

**No contraceptive-use at last sexual encounter.** Just over half (54%) of participants reported using contraception at their last sexual encounter which involved penetrative sex. Amongst those who used contraception, condoms were the most frequently used (37%), followed by a LARC method (14%) and contraceptive pills (7%).

In the univariate analysis, both a history of ecstasy use (OR 3.571, p = 0.002), being drunk or high at their last sexual encounter (OR 2.857, p = 0.009) and screening positive for either current PTSD (OR 2.497, p = 0.028) or current depression (OR 2.983, p = 0.005) were all associated with non-contraceptive use. However, following adjustment in the multivariate analysis, only ecstasy use (aOR 4.562, p = 0.004) remained significant. In addition, in the multivariate analysis, low scores on the sexual health knowledge test (aOR 3.706, p = .026) were significantly associated with non-contraceptive use. A history of physical abuse (aOR 3.024, p = 0.041) was also positively associated with contraception use (Table 4).

**LARC use.** Fourteen percent of participants reported use of a LARC method at their last sexual encounter. Not being drunk at their last sexual encounter was strongly associated with LARC use in both univariable (OR 0.214, p = 0.048) and multivariate analysis (aOR 0.146, p = 0.029). Again, a history of physical abuse (aOR 4.892, p = 0.050) was significantly associated with LARC use in the multivariate analysis with 95% confidence intervals ranging from 1.001 to 23.894. Identifying as queer, bi-sexual, lesbian or another non-heterosexual identity was associated with LARC use in the univariate analysis only (OR 3.2, p = 0.037

## Discussion

This study examined factors associated with both unintended pregnancy and contraceptive use in a justice-involved population of adolescent girls in Australia. Our population reported high levels of trauma histories, substance use, mental health problems and relatively low engagement with sexual health services compared to the general population, consistent with previous studies on justice-involved adult women [106–109]. Participants reported high levels of unintended pregnancy (26%) and subsequent miscarriage (60%) or termination (27%), as well as low levels of contraception usage (54%) compared to Australian adolescents in the community however use of a LARC method (14%) was significantly higher [14, 110]. This suggests a distinctive reproductive health profile to adolescents in the community who are not justice-involved.

The only factor independently associated with both an increased odds of unintended pregnancy and reduced odds of contraception-use at last sexual encounter in the multivariate analysis was a history of ecstasy use. This was an unexpected finding as we might have expected to observe a similar impact from other drugs such as alcohol, cannabis, methamphetamine and other amphetamines as indicated in past literature that has examined associations between substance use, unintended pregnancy and low contraceptive use [111–115]. A number of factors might have contributed to this finding. Firstly, the almost ubiquitous reported use of cannabis (87%) made it unlikely to find an association within this cohort as we lacked a sufficient sample of non-cannabis users with which to test association. Secondly, there may be differences in the profile of participants who take ecstasy (a largely non-addictive recreational drug) compared to more addictive substances such as methamphetamine and other amphetamines, which were not measured in this survey. This could relate to ecstasy being used as a "party drug", and being a more socially acceptable and accessible drug which appeals to people with low-risk perception [116], high levels of impulsivity [117] and low levels of self-control [118]. Such characteristics may transfer into similar attitudes around pregnancy and STI risk. Attention to this should be considered in future research and possibly by clinicians when providing counselling services to adolescents who report ecstasy use.

**Table 4. Factors associated with non-contraceptive use and LARC use at last sexual encounter among justice involved girls aged 14–17 years (n = 118).**

| Variable | | Non-contraception use | | | | LARC use | | | |
|---|---|---|---|---|---|---|---|---|---|
| | | Univariable OR (95% CI) | Reported P-value | Multivariable OR (95% CI) | Reported P-value | Univariable OR (95% CI) | Reported P-value | Multivariable OR (95% CI) | Reported P-value |
| Demographics | Location | | | | | | | | |
| | Urban | 1.0 | | 1.0 | | 1.0 | | 1.0 | |
| | Regional | 0.8 (0.3–2.0) | 0.7 | 0.7 (0.2–2.0) | 0.5 | 1.2 (0.4–4.2) | 0.7 | 0.9 (0.2–3.9) | 0.9 |
| | Aboriginal & Torres Strait Islander | | | | | | | | |
| | Yes | 1.0 | | 1.0 | | 1.0 | | 1.0 | |
| | No | 1.3 (0.6–2.6) | 0.5 | 1.3 (0.5–3.2) | 0.6 | 1.0 (0.4–2.9) | 0.9 | 0.6 (0.2–2.2) | 0.5 |
| | Unemployed or permanent disability | | | | | | | | |
| | No | 1.0 | | 1.0 | | 1.0 | | 1.0 | |
| | Yes | 1.1 (0.5–2.4) | 0.8 | 0.9 (0.4–3.2) | 0.8 | 0.7 (0.2–2.2) | 0.5 | 0.4 (0.1–1.6) | 0.2 |
| | Queer, Bi-sexual, Lesbian or other non-heterosexual identity | | | | | | | | |
| | No | 1.0 | | 1.0 | | 1.0 | | 1.0 | |
| | Yes | 0.9 (0.4–2.3) | 0.9 | 1.2 (0.4–3.6) | 0.8 | 3.2 (1.1–9.5) | 0.04 | 1.4 (0.3–5.4) | 0.7 |
| Abuse & sexual histories | Forced or frightened into sex | | | | | | | | |
| | No | 1.0 | | 1.0 | | 1.0 | | 1.0 | |
| | Yes | 1.2 (0.6–2.6) | 0.7 | 1.3 (0.4–3.8) | 0.7 | 2.0 (0.7–5.7) | 0.2 | 1.0 (0.2–4.5) | 0.9 |
| | Physical abuse | | | | | | | | |
| | No | 1.0 | | 1.0 | | 1.0 | | 1.0 | |
| | Yes | 0.8 (0.4–1.7) | 0.5 | 0.3 (0.1–1.0) | 0.04 | 3.6 (1.0–13.3) | 0.06 | 4.8 (1.0–23.9) | 0.05 |
| | Age difference at first sexual encounter ≥ 5 years | | | | | | | | |
| | No | 1.0 | | 1.0 | | 1.0 | | 1.0 | |
| | Yes | 1.4 (0.5–4.6) | 0.5 | 1.0 (0.2–4.6) | 1.0 | 3.1 (0.8–11.7) | 0.09 | 3.8 (0.6–24.6) | 0.2 |
| Mental Health | PTSD | | | | | | | | |
| | No | 1.0 | | 1.0 | | 1.0 | | 1.0 | |
| | Yes | 2.5 (1.1–5.7) | 0.03 | 2.3 (0.8–6.7) | 0.1 | 0.5 (0.2–1.5) | 0.5 | 1.0 (0.2–5.2) | 1.0 |
| | Past history of depressive episode | | | | | | | | |
| | No | 1.0 | | 1.0 | | 1.0 | | 1.0 | |
| | Yes | 3.0 (1.4–6.4) | 0.005 | 2.2 (0.8–6.0) | 0.1 | 0.5 (0.2–1.5) | 0.5 | 0.3 (0.1–1.4) | 0.1 |
| | Suicide attempt ever | | | | | | | | |
| | No | 1.0 | | 1.0 | | 1.0 | | 1.0 | |
| | Yes | 1.7 (1.6–0.8) | 0.2 | 1.0 (0.4–2.7) | 0.9 | 1.8 (0.6–5.0) | 0.3 | 1.8 (0.4–7.1) | 0.4 |
| Substance Use | Ecstasy | | | | | | | | |
| | No | 1.0 | | 1.0 | | 1.0 | | 1.0 | |
| | Yes | 3.6 (1.6–7.9) | 0.002 | 4.6 (1.6–12.8) | 0.004 | 1.0 (0.4–3.0) | 1.0 | 1.3 (0.3–5.0) | 0.7 |
| | Drunk or high at last sexual encounter | | | | | | | | |
| | No | 1.0 | | 1.0 | | 1.0 | | 1.0 | |
| | Yes | 2.9 (1.3–6.3) | 0.009 | 1.0 (1.0–1.0) | 0.8 | 0.2 (0.0–1.0) | 0.05 | 0.1 (0.0–0.8) | 0.03 |
| Other Factors | Scores on sexual health knowledge test | | | | | | | | |
| | High | 1.0 | | 1.0 | | 1.0 | | 1.0 | |
| | Low | 2.3 (0.9–5.7) | 0.09 | 3.7 (1.2–11.9) | 0.03 | 0.6 (0.2–2.0) | 0.4 | 0.7 (0.2–3.2) | 0.7 |

OR, odds ratio; CI, confidence interval

Alternatively, another factor to consider is the role of anxiety as a known risk factor for ecstasy use [119–122]. This, combined with anxiety acting as a pre-curser for poor contraception-negotiation skills [123–125] may exacerbate the risks for this population however this still does not explain why past ecstasy-use alone (as opposed to other substances) is an independent

risk factor for unintended pregnancy and non-contraception use given that anxiety is also a risk factor for cannabis, amphetamine and methamphetamine use [126, 127]. Finally, the relationship between ecstasy-use and risky sexual behaviour, caused by a combination of impaired cognitive functioning and heightened sexual arousal is well documented [128–132] however, in this sample, nearly half (49%) of all ecstasy users in the total sample reported only using ecstasy a few times a year, making it difficult to directly associate the use of the drug alone with an increased risk of unintended pregnancy and non-contraception use. It is most likely that the explanation for this finding involves a complex mix of the above-mentioned factors; further investigation into this finding is required.

Our study had some other unexpected findings. Firstly, a history of physical violence appeared to be associated with contraception use, including LARC use. This contradicts a relatively large body of work which shows a strong association between a history of physical trauma (including domestic violence) and non-contraception use [133–135]. Findings to be reported in a forthcoming publication of ours suggest that this may be due to the fact that many girls who reported a history of physical violence have also been placed in out of home care, and consequently, under the care of social workers. Our findings suggest that many social workers take a highly proactive role in facilitating access to contraception, making such adolescent girls more likely to be using LARC methods. Our survey did not collect information on out of home care histories, nor did it ask to differentiate between who the perpetrators of the physical abuse were (family vs partner, etc.) so this interpretation cannot be supported by the present study alone. We do however, recommend future studies consider the role out of home care and social workers may have in reproductive and other health outcomes for this population. Our study was also unable to distinguish between the types of physical violence experienced. For example, girls currently involved in physically violent relationships may be more likely to want to avoid pregnancy or seek out contraception which requires no partner negotiation, hence the overall higher rates of LARC-use observed in this group [136]. Future studies should consider such distinctions to allow for a more nuanced understanding of the factors associated with contraception use. Finally, while a statistically significant association was detected, the 95% confidence intervals, particularly for LARC use, were wide (1.001 to 23.894). Testing this association in a larger sample should be considered, as well as considerations for more upstream interventions of how to reduce exposure to violence.

Secondly, we found no association with demographic factors such as being unemployed or enrolled in school, being Aboriginal &/or Torres Strait Islander or living in a remote area, all of which have previously been associated with unintended pregnancies and poor contraception use [48, 60, 83, 137]. We further examined these characteristics by considering whether or not the two notable factors, ecstasy use and history of physical abuse, were significantly differently across different demographic characteristics by conducting chi-squared tests. With the exception of ecstasy use and Aboriginal & Torres strait Islander background, where 25% of Aboriginal & Torres strait Islander participants reported ecstasy use compared to 42% of non-Aboriginal participants (p = 0.042), there were no significant differences between other demographic characteristics and these two risk factors. This suggests that it is not these individual demographic characteristics alone which are driving unintended pregnancies and/or low contraception use among this demographic.

Finally, we also noted that participants who reported being intoxicated at their last sexual encounter were both more likely to not be currently using contraception and using LARC. This may form part of the story that we noted in the associations we observed with a history of physical violence, where girls engaged in high-risk behaviours were more likely to be identified as suitable candidates for LARC (by social workers, healthcare providers or other caretakers) due to their likelihood to not use contraception at all otherwise [138, 139].

Our findings should be considered in light of study limitations. Firstly, we have used the terminology of "unintended" pregnancy to reflect the questions asked of participants (ie. *"Was this a planned pregnancy? i.e. You wanted to get pregnant. It wasn't a surprise"),* and we recognise that the concept is disputable. The use of dichotomised terms as "planned/intended" vs "unplanned/unintended" has been criticised for over-simplifying a woman's fertility intentions, which are often a complex and nuanced based on a mixture of emotional, cognitive, social and structural dimensions [140, 141]. Secondly, the concept of "planning" a pregnancy is a relatively modern concept emanating from middle class constructs around the need to time and limit pregnancies for economic and health reasons [140]. While such ideas have merit, the notion of "planning" and "control" over a pregnancy is not universal to the lived experience of many women [140–142]. This has been particularly noted in the Australian context when examining the pregnancy intentions of Aboriginal women living in remote areas that appear to have very high levels of pregnancy ambivalence [60] as well as amongst low-income women in the US [143]. While public health policies almost universally advocate against teenage pregnancies due to poor health, socio-economic and psychological outcomes for young mothers and their children, this does not mean that all teenage pregnancies are viewed negatively by the adolescent girls who become pregnant or by their families/significant others [144]. It can be, in fact, viewed as a life-changing event that increases motivation for positive life changes and future life planning [7, 145]. Furthermore, there is acknowledgement within the literature that many of the adverse physical outcomes associated with teenage pregnancy are more a result of pre-existing social, economic and behavioural factors, as well as the stigma associated with teenage pregnancy [146], rather than specific adolescent biology [5] which contributes to these poor outcomes. Future studies need to ensure that they firstly, consider these dimensions properly and explore un-met contraceptive needs in the context of positive, ambivalent and negative attitudes towards current and past pregnancies [147, 148]; and secondly, consider how young adolescents who chose to become mothers can be better supported to avoid the adverse outcomes frequently reported in the literature.

We were only able to examine factor association in a relatively small sample of participants (n = 118) with small numbers of cases, particularly for unintended pregnancy (n = 28) and LARC use (n = 17), leading to a small number of independent variables per cases examined (<10). Consequently, we may have been unable to detect an effect from all independent variables examined [149]. Studies with larger sample sizes should be considered in the future, however, we believe that the findings of this small sample of participants is of importance given the limited amount of data available for this demographic.

Data collected in this study relied on self-report, which, when discussing trauma histories, risky behaviours and/or substance use may lead to under-reporting. Never-the-less when participants were asked at the end of the survey how many responses they believed they answered honestly; 94% responded with "all" or "most" increasing the reliability of the data. We are also unable to place the occurrence of events reported by participants within an appropriate timeline, consequently it is difficult to establish causation or determine which variables are causing the outcomes of interest. This is particularly true for associations with unintended pregnancy. We cannot say with certainty whether ecstasy use or the existence of PTSD occurred prior to or after the reported pregnancies. What we can establish, however, is that there appears to be an association between the two which warrants further investigation. We would also urge caution with the interpretation of screening positive for a mental health condition such as depression and/or PTSD as these scales have been designed to indicate a likelihood of a mental health condition; a true diagnosis is only possible following a clinical assessment which was not provided here.

While a multi-variate analysis approach was applied to better understand key associations and how they may interact, such models have their limitations in understanding what may actually drive adoption or non-adoption of a contraception method. Specifically in relation to LARC, we are unable to assess and how and by which mechanism LARC methods were provided to participants, the degree of consent and coercion and/or the degree to which this was a preferred method. Understanding these mechanisms in light of some of the concerns expressed about LARC promotion in minority populations is critical [32].

Finally, the purposive sampling approach was applied slightly differently in QLD and WA, meaning that this is not a truly representative sample. However, as stated above, the demographic profile appears consistent with the known demographic profile of this population.

## Conclusion

Our findings suggest that justice-involved adolescent girls have distinct reproductive health needs compared to the general population, with more research required to understand how these needs interact on the pathway to both non-contraception use and unplanned pregnancy. The high rates of unintended pregnancy, substance use, histories of sexual assault and other trauma histories within this groups highlights the need to consider more upstream determinants of violence and substance use within this cohort, while also re-considering how best to support such patients with shared-decision making approached when considering contraception options. Specifically, how exposure to physical violence may encourage contraception and LARC use, requires further investigation, as does ecstasy use.

## Supporting information

**S1 File. JOSH-MEH survey.**
(DOCX)

## Acknowledgments

We would like to thank all the participants of the MeH-JOSH study for their participation and engagement with this study.

## Author Contributions

**Conceptualization:** Helene Smith, Tony Butler, Sally Nathan.

**Formal analysis:** Helene Smith.

**Funding acquisition:** Tony Butler.

**Methodology:** Tony Butler, Sally Nathan.

**Project administration:** Tony Butler.

**Supervision:** Mandy Wilson, Basil Donovan, Jocelyn Jones, Paul Simpson.

**Writing – original draft:** Helene Smith.

**Writing – review & editing:** Mandy Wilson, Basil Donovan, Jocelyn Jones, Tony Butler, Sally Nathan, Paul Simpson.

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
