## [Decision Letter · Decision Letter 0]

28 Jun 2023

PONE-D-23-13119Factors associated with unintended pregnancy and contraceptive practices in justice-involved adolescent girls in Australia.PLOS ONE

Dear Dr. Smith,

Thank you for submitting your manuscript to PLOS ONE. After careful consideration, we feel that it has merit but does not fully meet PLOS ONE’s publication criteria as it currently stands. Therefore, we invite you to submit a revised version of the manuscript that addresses the points raised during the review process. While some of the reviewers were more positive, I am in agreement with the reviewers overall that substantial changes are required to render the manuscript acceptable for publication. Please attend to all of the points raised by the reviewers.

We look forward to receiving your revised manuscript.

Kind regards,

Andrea Knittel

Academic Editor

PLOS ONE

2. 1) You indicated that you had ethical approval for your study. In your Methods section, please ensure you have also stated whether you obtained consent from parents or guardians of the minors included in the study or whether the research ethics committee or IRB specifically waived the need for their consent.

    2) Please provide additional information regarding the considerations  made for the prisoners included in this study. For instance, please discuss whether participants were able to opt out of the study and whether individuals who did not participate receive the same treatment offered to participants.

Reviewers' comments:

Reviewer's Responses to Questions

**Comments to the Author**

1. Is the manuscript technically sound, and do the data support the conclusions?

Reviewer #1: Partly

Reviewer #2: Yes

Reviewer #3: Partly

2. Has the statistical analysis been performed appropriately and rigorously? 

Reviewer #1: I Don't Know

Reviewer #2: Yes

Reviewer #3: I Don't Know

3. Have the authors made all data underlying the findings in their manuscript fully available?

Reviewer #1: No

Reviewer #2: Yes

Reviewer #3: Yes

4. Is the manuscript presented in an intelligible fashion and written in standard English?

Reviewer #1: Yes

Reviewer #2: Yes

Reviewer #3: Yes

5. Review Comments to the Author

Reviewer #1: This is an important study that uses unique data from a vulnerable population in an ethical manner. Overall it is well written and the results are appropriately contextualized. My main suggestions are the following:

- In the abstract and throughout the main body of the manuscript, I recommend prioritizing the N instead of the %. This is because the total sample size is relatively small and close to 100. When is space to report both N and percentage, report both. If there's only space to report one number, I recommend reporting N.

- I've recommended the editors invite a statistical reviewer to evaluate the use of multivariate analyses in this study. I believe that the authors should detail exactly which independent variables were included in each model. In addition, I question the identification of independent variables using a p-value threshold instead of (for instance) a priori based on theoretical grounds.

Reviewer #2: The authors present a study examining factors associated with unintended pregnancy, non-contraceptive use and Long-Acting Reversible Contraception (LARC) in a sample of sexually active, justice-involved adolescent girls from Western Australia and Queensland. I though the authors did an excellent job outlining the information in a clear and succinct manner.

Comments

Introduction: The authors clearly described the background and significance of this issue, not only nationally (Australia), but also internationally. I was able to walk away with a clear understanding of the population (why aboriginal/strait islanders where used as the reference population), as well as, why LARC was the contraceptive method of interest.

Methods: The methods section was clear and is easily reproducible. The only suggestion I would offer is to more explicitly state the study design type.

Results: Minor style recommendation-I recommend reporting all data with consistent decimal points (i.e. data in tables rounded to 1 decimal place, data in text variable)

Discussion: This section appropriately addressed many of the questions that arose from the result section. I would have liked to see a little more discussion around the factors associated with LARC use, notably "being drunk at the last sexual encounter" and sexual orientation. Those results were surprising to me and contextualizing it in the discussion section would have been helpful.

Overall, I thought this study was well done and warrants acceptance.

Reviewer #3: Thank you for the opportunity to review this manuscript “Factors associated with unintended pregnancy and contraceptive practices in justice-involved adolescent girls in Australia.” This is an important topic and addresses issues of health inequities in an often overlooked population. The manuscript would benefit from revisions and clarifications. In particular, there is an unquestioned assumption that adolescents must be on LARCs to prevent unintended pregnancy, but there is some nuance there, rooted in colonial histories of oppression and eugenicist ideas of stratified reproduction, that should be thought through more (I’ve provided suggestions on where), and missing information about what happened to the unintended pregnancies. It is not simply the fact of unintended pregnancy that may be associated with certain negative outcomes, but whether people parent or not.

More detailed comments are below.

Page 2

Abstract

Line 33 and 44 and main text- recommend consistent language, as adolescent girls vs. young girls are both used here, and at other places you use adolescent women. Please choose one and use throughout.

Methods needs a little more detail as to what was done/study design, not just what database was used and the sampling, but type of study design and outcomes measured.

Line 33- comma after girls and before who

Lines 78-86. This discussion of LARC would benefit from being balanced by the critique of these campaigns to promote provider-controlled LARC methods in adolescents (and other groups) over other reversible methods. Please explain why you prioritize LARC as something that should be encouraged. While there has been an assumption that people, especially adolescents, should use these highly effective methods, this assumption has also been shown to be laden with pejorative judgements about whose reproductive capacities are more valued and whose should be suppressed—see for instance Gomez AM “Women or LARC first?” Perspect Sex Reprod Health. 2014.

Line 95 and 127- Please define reproductive coercion as many readers may not be familiar with the term. In addition, there are (at least) 2 levels of reproductive coercion- one at the partner level, and one at the systemic level with providers coercing patients into choosing certain methods; this latter point is relevant for and has been documented in the literature for LARC methods, and therefore needs some mention and teasing out.

Paragraph beginning line 99- Please provide some information on how many adolescent females are involved in the justice system.

139 This sentence would benefit from a little tweaking for clarity to sign post for readers that you are now shifting into describing what you are doing in the present study. A broad statement of the type of study design would be helpful

142-143- This kind of a statement is better in the discussion. In the intro, you can set this idea up earlier by noting that this is a gap in the literature.

155- please avoid the use of the term “offender” in favor of humanizing language.

For readers unfamiliar with the Australian juvenile carceral system and access to health care, it would be very helpful in the intro to provide some “setting” background on what the status of access to health care is like, and whether contraception access in custody is routine (perhaps it is in here and I missed it?). Relatedly, readers will want to know what LARC access and use among adolescents in Australia is like for context to the study population/research question.

Methods

Before describing the sampling, it would be helpful to explain what kind of data were collected—the fact that you interviewed participants via survey doesn’t come through until 183, and even then you assume readers know that this study involved surveying people (“the survey was delivered”) but you haven’t yet explained that the overall study design was a cross sectional survey. This would be helpful to clarify earlier on, and then you can get into the details of sampling, recruitment, survey design and administration.

156- comma after Islander girls. Also, how did you determine their identity category? Self identified?

215- Unintended pregnancy variable- did you collect data on what happened to those pregnancies? Abortion? Birth? Miscarriage? And if birth, parenting or adoption? This is important information for the context of the research questions you have set out to know, especially since you the frame the study so much in terms of the harms of unintended pregnancy. If this was not collected, then this does need justification. Otherwise, please report.

218-219- The list of what was included as a modern contraceptive method excludes some commonly used modern methods. Why does this not include contraceptive patch, contraceptive vaginal ring, injectable contraception, and IUDs?

In the outcomes in the abstract at least you report ‘contraceptive use,’ not just non-contraceptive use. Does contraceptive use include LARC? If not, then you should rename the category to be specific to what you have included.

222- The definition of what the authors are including as a long-acting reversible contraceptive is not standard. On lines 81-82, the authors DO use the standard definition, so it is confusing here that they have changed the categorization. LARCs include intrauterine devices and subdermal contraceptive implants. Injectable is typically not considered a LARC- if the authors are using a particular organization’s categorizations of LARCs that includes injectable, please cite it. Or if the authors have another compelling reason why they want to include injectable in this category of outcomes, then the category should be renamed to signal it is inclusive of injectable; otherwise, the authors would need to re-do analyses to exclude injectable from the LARC category.

223- does ‘implant’ here refer to subdermal contraceptive implants or other intrauterine implants?

Furthermore, why are copper IUDs not included? And recommend not referring to “Mirena” by its brand name, but rather hormonal or levonorgestrel released IUD (per WHO nomenclature guidance).

Results

266-307- This is a lot of text to describe what is in Table 1. Suggest streamlining so as to not duplicate too much what readers can learn from looking at Table 1.

347- not being drunk at last sexual encounter being associated with LARC use will need to be teased out in the discussion. Use of LARC (and any specific choice of method) depends on so many factors, and with LARC in particular it is highly dependent on structural factors of access. How would being drunk relate to whether a community provides youth access to LARC?

Discussion

The discussion would benefit from a greater consideration of issues of LARC access (as mentioned above) as one factor determining use of LARC, both in the broader discussion and in the limitations.

As with the introduction, a consideration of the nuance of LARC access in youth, without taking for granted the assumption that the goals is for more youths to use LARCs (see Gomez article). What about goals of reducing violence and other upstream factors as a takeaway from this study?

Also the issue of what happened to those unintended pregnancies must be addressed. If the authors did not collect this at all, this is a major limitation.

The discussion of the nuance of the categories unplanned/unintended is well done.

496-498- The authors could make a stronger conclusion oriented toward violence prevention and treatment for the sequellae, rather than focusing solely on the goal of increasing LARC use. The authors could be more specific about “further attention”—they have enough compelling data to call for action, policy and service recommendations rather than a generic “further attention.”

6. PLOS authors have the option to publish the peer review history of their article (what does this mean?). If published, this will include your full peer review and any attached files.

Reviewer #1: No

Reviewer #2: No

Reviewer #3: No

---

## [Author Response · Author response to Decision Letter 0]

29 Jan 2024

General PLOS ONE Feedback

1. “Please ensure that your manuscript meets PLOS ONE's style requirements, including those for file naming. The PLOS ONE style templates can be found at https://journals.plos.org/plosone/s/file?id=wjVg/PLOSOne_formatting_sample_main_body.pdf _ and https://journals.plos.org/plosone/s/file?id=ba62/PLOSOne_formatting_sample_title_authors_affiliations.pdf”

Thank-you for pointing this out. We have reviewed the two links provided in detail and have made the necessary edits including heading titles, labelling of author contributions, font sizes and referencing. 

2. “You indicated that you had ethical approval for your study. In your Methods section, please ensure you have also stated whether you obtained consent from parents or guardians of the minors included in the study or whether the research ethics committee or IRB specifically waived the need for their consent.”

Thank-you for asking for clarification on this important issue. Please note that in the section entitled Consent [LINES 303-311], we outlined that a waiver of consent was provided by the research ethics committee, however, where possible, we also sought parental consent. We have slightly clarified the language to ensure clarity on this issue [LINE 305]. 

3. “Please provide additional information regarding the considerations made for the prisoners included in this study. For instance, please discuss whether participants were able to opt out of the study and whether individuals who did not participate receive the same treatment offered to participants.”

We would like to clarify that while many participants had a history of incarceration, no participants were actively being detained or serving a sentence when they were recruited and participated in the study. We have clarified this point to ensure there is no confusion. [LINE 276-277]. 

4. “We note that you have indicated that data from this study are available upon request. PLOS only allows data to be available upon request if there are legal or ethical restrictions on sharing data publicly. For more information on unacceptable

 data access restrictions, please see http://journals.plos.org/plosone/s/data-availability#loc-unacceptable-data-access-restrictions.

If there are ethical or legal restrictions on sharing a de-identified data set, please explain them in detail (e.g., data contain potentially sensitive information, data are owned by a third-party organization, etc.) and who has imposed them (e.g., an ethics committee). Please also provide contact information for a data access committee, ethics committee, or other institutional body to which data requests may be sent.

If there are no restrictions, please upload the minimal anonymized data set necessary to replicate your study findings as either Supporting Information files or to a stable, public repository and provide us with the relevant URLs, DOIs, or accession numbers. For a list of acceptable repositories, please see http://journals.plos.org/plosone/s/data-availability#loc-recommended-repositories.

We will update your Data Availability statement on your behalf to reflect the information you provide.”

Thank-you for asking us to clarify this important issue. We are unable to make the data available publicly for a number for reasons related to the ethical restrictions placed by our ethics research committee when providing consent for this study. 

i. The data set contains highly sensitive information about participants’ sexual history, past substance use, medical history (including HIV status, history of mental health), criminal activity (such as illegal drug use and paid sex work), and reproductive health history (including history of terminating pregnancies). In addition, it includes histories of sexual and physical assault, which may or may not have been previously disclosed to the police. Linking of any of these issues to a specific participant could be highly detrimental to the physical and mental wellbeing of the participant, as potentially expose the participant to both stigma and/or potential legal complications. 

ii. Whilst all data has been de-identified, the relatively small sample size (118), combined with geographical information, details of criminal history, age of sexual partners and other identifying information, could make a participant identifiable, or (just as damagingly) have an individual be perceived to be a specific participant who has engaged in certain stigmatising behaviour. Given the high level of vulnerability of this cohort, many of which may be waiting on sentencing, parole orders, custody arrangements or other interactions with government and justice bodies, we do not believe the risk of disclosure to be acceptable by making such a dataset publicly available.

iii. As part of the consent process, participants were assured that their data would remain private and only be made available with the express permission of the investigator team where we could ensure the ongoing privacy of the data. We therefore believe that making this data (even if de-identified) publicly available would be a breach of our agreement with participants. 

We are, however, able to provide the data set for review by individuals at PLOS ONE upon request. The contact details for the ethics committee are listed below:

NSW Human Research Ethics Committee

HREC Ref: # HC13308

humanethics@unsw.edu.au

02 9348 1943 

5. “PLOS requires an ORCID ID for the corresponding author in Editorial Manager on papers submitted after December 6th, 2016. Please ensure that you have an ORCID ID and that it is validated in Editorial Manager. To do this, go to ‘Update my Information’ (in the upper left-hand corner of the main menu), and click on the Fetch/Validate link next to the ORCID field. This will take you to the ORCID site and allow you to create a new ID or authenticate a pre-existing ID in Editorial Manager. Please see the following video for instructions on linking an ORCID ID to your Editorial Manager account: https://www.youtube.com/watch?v=_xcclfuvtxQ “

We have provided the corresponding author’s ORCID ID Number (0000-0001-7456-0914) and have updated in the Editorial Manager. 

Comments from Reviewer #1

6. This is an important study that uses unique data from a vulnerable population in an ethical manner. Overall, it is well written, and the results are appropriately contextualized. My main suggestions are the following:

“In the abstract and throughout the main body of the manuscript, I recommend prioritizing the N instead of the %. This is because the total sample size is relatively small and close to 100. When is space to report both N and percentage, report both. If there's only space to report one number, I recommend reporting N.”

Thank-you for making this observation. We have made these edits throughout the paper. 

7. I've recommended the editors invite a statistical reviewer to evaluate the use of multivariate analyses in this study. I believe that the authors should detail exactly which independent variables were included in each model. In addition, I question the identification of independent variables using a p-value threshold instead of (for instance) a priori based on theoretical grounds.

We welcome the addition of a statistical reviewer, in the meantime, we have provided additional information of which variables were included in each model alongside a rationale for their inclusion [LINE 382-395]. 

Comments from Reviewer #2

8. The authors present a study examining factors associated with unintended pregnancy, non-contraceptive use and Long-Acting Reversible Contraception (LARC) in a sample of sexually active, justice-involved adolescent girls from Western Australia and Queensland. I though the authors did an excellent job outlining the information in a clear and succinct manner.

Introduction: The authors clearly described the background and significance of this issue, not only nationally (Australia), but also internationally. I was able to walk away with a clear understanding of the population (why aboriginal/strait islanders were used as the reference population), as well as, why LARC was the contraceptive method of interest.

Methods: The methods section was clear and is easily reproducible. The only suggestion I would offer is to more explicitly state the study design type.”

Thank-you for the encouraging feedback. We have added in a paragraph at the start of the methods to outline study design, including specifying that this was a cross-sectional study. [LINE 265-272]

9. “Results: Minor style recommendation-I recommend reporting all data with consistent decimal points (i.e. data in tables rounded to 1 decimal place, data in text variable)

have done to 0 decimal places for consistency.”

Thank-you for this observation. We have ensured that all descriptive data is reported as a whole number and have removed the decimal places from Tables 1 & 2. [LINES 417 & 516]

10. “Discussion: This section appropriately addressed many of the questions that arose from the result section. I would have liked to see a little more discussion around the factors associated with LARC use, notably "being drunk at the last sexual encounter" and sexual orientation. Those results were surprising to me and contextualizing it in the discussion section would have been helpful”.

Thank-you for this observation, we have added two paragraphs in the discussion that address considerations around the results relating to LGBT participants and being drunk at last sexual encounter. [LINES 685 - 706]

Comments from Reviewer #3

11. “Thank you for the opportunity to review this manuscript “Factors associated with unintended pregnancy and contraceptive practices in justice-involved adolescent girls in Australia.” This is an important topic and addresses issues of health inequities in an often-overlooked population. The manuscript would benefit from revisions and clarifications. In particular, there is an unquestioned assumption that adolescents must be on LARCs to prevent unintended pregnancy, but there is some nuance there, rooted in colonial histories of oppression and eugenicist ideas of stratified reproduction, that should be thought through more (I’ve provided suggestions on where), and missing information about what happened to the unintended pregnancies. 

It is not simply the fact of unintended pregnancy that may be associated with certain negative outcomes, but whether people parent or not.

More detailed comments are below.

We thank-you for pointing out this over-riding comment and fully agree that more space is needed to discuss the nuances of this issue. We addressed your individual comments below. 

12. Abstract: Line 33 and 44 and main text- recommend consistent language, as adolescent girls vs. young girls are both used here, and at other places you use adolescent women. Please choose one and use throughout”.

We have reviewed the abstract adjusted our language to say adolescent girls throughout [LINE 69]. This term is now used consistently throughout the text. 

13. “Methods needs a little more detail as to what was done/study design, not just what database was used and the sampling, but type of study design and outcomes measured”.

Thank-you for this comment. We have provided additional information speaking to the study design and outcome variables (while remaining succinct to stick to the word limit) [LINES 70-71

14. “Line 33- comma after girls and before who”

Thank-you for picking this up. We have added [LINE 58]

15. “Lines 78-86. This discussion of LARC would benefit from being balanced by the critique of these campaigns to promote provider controlled LARC methods in adolescents (and other groups) over other reversible methods. Please explain why you prioritize LARC as something that should be encouraged. While there has been an assumption that people, especially adolescents, should use these highly effective methods, this assumption has also been shown to be laden with pejorative judgements about whose reproductive capacities are more valued and whose should be suppressed—see for instance Gomez AM “Women or LARC first?” Perspect Sex Reprod Health. 2014”.

Thank-you for highlighting the need to present some of the frictions that exist within pro-LARC policy recommendations. We fully agree and have added in two additional paragraphs. The first outlines the reasons why LARC is recommended by several medical bodies for adolescent use [LINES 120 – 128], and the second outlines some of the rising concerns that have begun to emerge with its use, particularly in the context of justice-involved women. This includes reference to the article by Gomez 2014, as you have suggested. [LINES 157 - 172]

16. “Line 95 and 127- Please define reproductive coercion as many readers may not be familiar with the term” In addition, there are (at least) 2 levels of reproductive coercion- one at the partner level and one at the systemic level with providers coercing patients into choosing certain methods; this latter point is relevant for and has been documented in the literature for LARC methods, and therefore needs some mention and teasing out”. 

Thank-you for this suggestion, we have provided a definition which encompasses both individual and structural instances of reproductive coercion [LINES 189 - 199]. 

17. Paragraph beginning line 99- Please provide some information on how many adolescent females are involved in the justice system.

This has been added [LINE 212 - 214]

18. Line 139 - This sentence would benefit from a little tweaking for clarity to sign post for readers that you are now shifting into describing what you are doing in the present study. A broad statement of the type of study design would be helpful

We have clarified [LINE 265 - 272]

19. 142-143- This kind of a statement is better in the discussion. In the intro, you can set this idea up earlier by noting that this is a gap in the literature.

Thank-you for this comment which we have taken on board by removing the general statement and outlining explicitly the type of analysis to be conducted. [LINE 256]

20. 155- please avoid the use of the term “offender” in favour of humanizing language.

Thank-you for picking this up. We have changed the word “offender” to Justice-involved and clarified a definition of what this involves. [LINE 159 - 161]

21. For readers unfamiliar with the Australian juvenile carceral system and access to health care, it would be very helpful in the intro to provide some “setting” background on what the status of access to health care is like, and whether contraception access in custody is routine (perhaps it is in here and I missed it?). Relatedly, readers will want to know what LARC access and use among adolescents in Australia is like for context to the study population/research question.

Thank-you for highlighting the importance of explaining the context for an international audience. We have addressed this issue in several places throughout the paper:

• Data outlining low uptake rate in the general population amongst adolescents [LINE 131]]

• Outline of how it is accessed and associated issues relating to costs [LINES 150 – 156] 

• Comparison to the community in the discussion [LINE 603]

Please note that, as outlined in point #3, all participants were interviewed in the community. Consequently, we have provided an outline of how contraception is accessed in the general population in Australia rather than within the incarceration setting.

22. Methods: Before describing the sampling, it would be helpful to explain what kind of data were collected—the fact that you interviewed participants via survey doesn’t come through until 183, and even then, you assume readers know that this study involved surveying people (“the survey was delivered”) but you haven’t yet explained that the overall study design was a cross sectional survey. This would be helpful to clarify earlier on, and then you can get into the details of sampling, recruitment, survey design and administration.

As addressed in comment #8, we have added in a paragraph at the start of the methods to outline study design, including specifying that this was a cross-sectional study. [LINE 265-272]

23. Line 156- comma after Islander girls. Also, how did you determine their i

---

## [Editor Report · Decision Letter 1]

5 Mar 2024

PONE-D-23-13119R1Factors associated with unintended pregnancy and contraceptive practices in justice-involved adolescent girls in Australia.PLOS ONE

Dear Dr. Smith,

Thank you for submitting your manuscript to PLOS ONE. After careful consideration, we feel that it has merit but does not fully meet PLOS ONE’s publication criteria as it currently stands. Therefore, we invite you to submit a revised version of the manuscript that addresses the points raised during the review process. I appreciate your responsiveness to the reviewer comments. I have identified a few minor issues that have arisen with the revision and have outlined those below.

We look forward to receiving your revised manuscript.

Kind regards,

Andrea Knittel

Academic Editor

PLOS ONE

Journal Requirements:

Additional Editor Comments:

Several small issues have been introduced in the process of revision. I have outlined these here:

Line 101: IUD = Intrauterine device (not inter-uterine). Please ensure this is correct throughout.

Line 148: The revisions suggested that all instances of “young women” or other terms had been changed to “adolescent girls.” Consider also changing in this instance.

Line 158: Reproductive coercion would be more appropriately frame as gender neutral. Consider: “Reproductive coercion refers to where a person is exposed to behavior interfering with the ability to make autonomous decisions about reproductive health. This could involve…force a person to either terminate or keep a pregnancy against their wishes.” And so on.

Line 304: The definition of “modern” contraceptive methods remain odd. It seems a stretch to describe condoms as modern and the phrase “short and long-acting” is strange. Consider instead removing the modifier “modern” and just listing the options that were included in their entirety.

Table 1: Many of the labels are unclear. For example, presumably “forced or frightened to have sex” means that this was done to the participants, not that they engaged in this behavior. It is not clear what “Last penetrative sex with regular partner” means. Please review these descriptors to ensure that the tables can stand alone.

Line 501: This is now a run-on sentence. Please revise.

Line 581-582: This sentence seems to confuse LGBTQ+ identity with behavior. If prior literature focuses on behavior, consider clarifying this, or use the “identity/identify” language throughout.

---

## [Author Response · Author response to Decision Letter 1]

22 Apr 2024

We have reviewed all feedback and made the recommended changes as requested. These are listed below, alongside a detailed outline of how each point has been addressed. 

Please note that the line references below refer to the line reference in the TRACKED CHANGES document. 

1. “Please review your reference list to ensure that it is complete and correct. If you have cited papers that have been retracted, please include the rationale for doing so in the manuscript text or remove these references and replace them with relevant current references. Any changes to the reference list should be mentioned in the rebuttal letter that accompanies your revised manuscript. If you need to cite a retracted article, indicate the article’s retracted status in the References list and also include a citation and full reference for the retraction notice.”

We have reviewed all references and made a few small amendments throughout. Should there be any additional concerns about any specific references, please do let us know. 

2. Line 101: IUD = Intrauterine device (not inter-uterine). Please ensure this is correct throughout.

Thank-you for picking this up. We have fixed the typo using the full term (intrauterine) and the abbreviation (IUD) on line 79 and then used the abbreviation only throughout the rest of the document. 

3. Line 148: The revisions suggested that all instances of “young women” or other terms had been changed to “adolescent girls.” Consider also changing in this instance.

Thank-you for this observation. This has now been amended so that “adolescent girls” is used throughout the manuscript [LINES 122 – 125 & 173]

4. Line 158: Reproductive coercion would be more appropriately frame as gender neutral. Consider: “Reproductive coercion refers to where a person is exposed to behaviour interfering with the ability to make autonomous decisions about reproductive health. This could involve…force a person to either terminate or keep a pregnancy against their wishes.” And so on.

Thank-you for this observation, we have adopted your suggested language to ensure that it is a more gender-neutral definition. [LINES 130 – 132]

5. Line 304: The definition of “modern” contraceptive methods remains odd. It seems a stretch to describe condoms as modern and the phrase “short and long acting” is strange. Consider instead removing the modifier “modern” and just listing the options that were included in their entirety.

Thank-you for this query. We have re-reviewed the literature and can confirm that the term modern does include condoms and is defined as “any “product or medical procedure that interferes with reproduction from acts of sexual intercourse”. This is opposed to what are considered more “traditional” or “natural” forms of contraception such as withdrawal or the rhythm method. To avoid confusion for any readers, we have clarified the definition and appropriate references within the text. [LINE 280 -284]. 

6. Table 1: Many of the labels are unclear. For example, presumably “forced or frightened to have sex” means that this was done to the participants, not that they engaged in this behaviour. It is not clear what “Last penetrative sex with regular partner” means. Please review these descriptors to ensure that the tables can stand alone.

Thank-you for your review of these labels and identifying certain labels that were not immediately clear. We have reviewed and amended a number of labels including “History of serious sentence”, “Types of offences previously committed”, “Abuse history inflicted on the participant”, “Past head injury”, “Has previously sought sexual health advice from a clinical professional” and “Last penetrative sexual encounter was with a regular partner” [LINE 347]

7. Line 501: This is now a run-on sentence. Please revise.

Please note that the line numbers that were provided in the feedback form did not seem to match the lines provided in either the clean or tracked version. In most instances we have been able to ascertain which section of the paper you were referring to, however in this instance we believe that you were referring to the sentence on LINES 499 – 502, which we have revised accordingly. Please do clarify if this was not the sentence that you were referring to. 

8. Line 581-582: This sentence seems to confuse LGBTQ+ identity with behaviour. If prior literature focuses on behaviour, consider clarifying this, or use the “identity/identify” language throughout.

Thank-you for your query and concern around this area. The authors have reviewed this and reconsidered our response and text inserted on this topic. We too were surprised to see the association in the univariate analysis between LARC use and sexual orientation because this is not supported by the literature. However, an important part of conducting a study like this is undertaking a multivariate analysis, in case there are confounding effects that we cannot see or identify in the univariate analysis. What we have found when we run this multivariate analysis, that adjusts for all other potential confounding variables, this association disappears and therefore is consistent with previous literature. Therefore, we believe that our focus should be on the multivariate analysis in the discussion. We include the univariate just to see what lost significance when we ran the multivariate analysis. Consequently, we have removed this section from the discussion.

---

## [Editor Report · Decision Letter 2]

20 May 2024

Factors associated with unintended pregnancy and contraceptive practices in justice-involved adolescent girls in Australia.

PONE-D-23-13119R2

Dear Dr. Smith,

We’re pleased to inform you that your manuscript has been judged scientifically suitable for publication and will be formally accepted for publication once it meets all outstanding technical requirements.

Kind regards,

Andrea Knittel

Academic Editor

PLOS ONE
---

## [Editor Report · Acceptance letter]

23 May 2024

PONE-D-23-13119R2 

PLOS ONE

Dear Dr. Smith, 

I'm pleased to inform you that your manuscript has been deemed suitable for publication in PLOS ONE. Congratulations! Your manuscript is now being handed over to our production team.

Kind regards, 

on behalf of

Dr. Andrea Knittel 

Academic Editor

PLOS ONE